# Do mothers who delivered at health facilities return to health facilities for postnatal care follow-up? A multilevel analysis of the 2016 Ethiopian Demographic and Health Survey

**Brhane Ayele**[1]*, **Mulugeta Woldu**[1], **Haftom Gebrehiwot**[2], **Tsegay Wellay**[2], **Tsegay Hadgu**[1], **Hailay Gebretnsae**[1], **Alemnesh Abrha**[1], **Equbay Gebre-egziabher**[1], **Sarah Hurlburt**[3]

**1** Tigray Health Research Institute, Mekelle, Tigray, Ethiopia, **2** College of Health Sciences, Mekelle University, Mekelle, Ethiopia, **3** Fenot, a Project of the Harvard T.H. Chan School of Public Health, Addis Ababa, Ethiopia

* brhane3127@gmail.com

## Abstract

### Introduction

Returning to health facility for postnatal care (PNC) use after giving birth at health facility could reflect the health seeking behavior of mothers. However, such studies are rare though they are critically important to develop vigorous strategies to improve PNC service utilization. Therefore, this study aimed to determine the magnitude and factors associated with returning to health facilities for PNC among mothers who delivered in Ethiopian health facilities after they were discharged.

### Methods

This cross-sectional study used 2016 Ethiopian Demographic and Health Survey data. A total of 2405mothers who gave birth in a health facility were included in this study. Multilevel mixed-effect logistic regression model was fitted to estimate both independent (fixed) effects of the explanatory variables and community-level (random) effects on return for PNC utilization. Variable with p-value of $\leq 0.25$ from unadjusted multilevel logistic regression were selected to develop three models and p-value of $\leq 0.05$ was used to declare significance of the explanatory variables on the outcome variable in the final (adjusted) model. Analysis was done using IBM SPSS statistics version 21.

### Result

In this analysis, from the total 2405 participants, 14.3% ((95%CI: 12.1–16.8), (n = 344)) of them returned to health facilities for PNC use after they gave birth at a health facility. From the multilevel logistic regression analysis, being employed (AOR = 1.51, 95%CI: 1.04–2.19), receiving eight and above antenatal care visits (AOR = 2.90, 95%CI: 1.05–8.00), caesarean section delivery (AOR = 2.53, 95%CI: 1.40–4.58) and rural residence (AOR = 0.56,

**Data Availability Statement:** All relevant data are within the paper and its Supporting Information files.

**Funding:** The authors received no specific funding for this work.

**Competing interests:** The authors have declared that no competing interests exist.

**Abbreviations:** ANC, Antenatal Care; AOR, Adjusted Odds Ratio; EDHS, Ethiopian Demographic Health Survey; PNC, Postnatal Care; WHO, World Health Organization.

95%CI: 0.36–0.88) were found significantly associated with return to health facilities for PNC use among women who gave birth at health facility.

## Conclusion

Facility-based PNC utilization among mothers who delivered at health facilities is low in Ethiopia. Both individual and community level variables were determined women to return to health facilities for PNC use. Thus, adopting context-specific strategies/policies could improve PNC utilization and should be paid a due focus.

## Introduction

Postnatal care service is an effective intervention to reduce morbidity and mortality of both mothers and newborns if it is given in a timely manner, with adequate frequency and including full service components [1]. Though maternal, newborn and child health issues are the national priorities in many countries, effective postpartum care implementation in developing countries in general and Sub-Saharan Africa in particular remains weak [2, 3].

The World Health Organization (WHO) recommends that a woman and her baby should be assessed by a health professional within one hour of birth, and again before discharge from a facility; especially for institutional births as opportunities are in place, this assessment could continue up to 24 hours after delivery which is a time seated for first contact of PNC [4]. The other follow-up contacts are recommended at 2–3 days, 6–7 days and 6 weeks (4 postnatal visits in total) [5–7].

Many studies in developing countries reported that mothers who delivered in a health facility were more likely to report attending postnatal care visit [1, 8–11]. However, even for deliveries at health facility level, PNC is a neglected service in Sub-Saharan African countries, where women are often discharged before 24 hours postpartum, which limits them from receiving the WHO's recommended services [3, 12]. In some studies, health facility delivery has been associated with lower postnatal care service utilization, with cited reasons including; mothers who delivered at the health facility were not advised when to return and more complications among mothers who delivered at home [12, 13].

While improvements have been made for Antenatal Care (ANC) use and skill birth attendance, PNC service utilization in Ethiopia remains low. The 2016 Ethiopian Demographic Health Survey (EDHS) result revealed that among women age 15–49 who gave birth, only 17% had a postnatal check during the first 2 days after birth, and four out of five women (81%) did not receive a postnatal check at all [14]. Other Ethiopian studies have also documented low (though higher than DHS) PNC service utilization, which ranges from 31.7% in Orromia to 65.6% in Addis Ababa [15–18]. The low utilization rates, and intra-setting difference of PNC rates show the real low coverage, but also the inconsistency in reporting practices. For example, health facilities reported that women who received immediate childbirth care before discharge at a facility were considered, by default, as having received postnatal care [12, 13], which may result in over reporting of PNC service utilization. On the other hand, studies on how many mothers are returned to health facilities for PNC use after they delivered at health facilities are rare. Four studies in Ethiopia further analyzed 2016-EDHS data to explore PNC service utilization considering different target populations: among fourth ANC utilizers [19], among home deliveries [20], and among home and health facility deliveries [21, 22].

Health facility based PNC use among women who delivered at health facilities (after they discharged from health facility for their facility based delivery) could reflect PNC seeking

behavior of mothers. However, no study was conducted on health facility based PNC after discharged for health facility based delivery and no analysis was carried out for the DHS data in Ethiopia though it is critically important to develop vigorous strategies to improve PNC service utilization. Therefore, this study aimed to determine the magnitude and factors associated with returning to health facilities for PNC use among mothers who delivered in Ethiopian health facilities after they were discharged.

## Methods

### Study design and data source

A cross-sectional study design using secondary analysis of 2016 Ethiopian Demographic and Health Survey data was used. The 2016 Ethiopian DHS data is the fourth series which was collected by the Central Statistical Agency (CSA), Ethiopian Public Health Institute (EPHI) and the DHS Program, International Classification of Functioning (ICF). To collect the data, two-stage stratified (urban and rural) sampling technique was employed in the survey to select Enumeration Areas (EAs) in the first stage and households in the second stage. Its further sampling technique is explained elsewhere [14].

The data for mothers who delivered at a health facility was extracted from the Individual Record (IR) dataset of the EDHS 2016. Only the most recent child-birth of the women was included in the analysis, to avoid mix-ups in the recall and reporting of mothers' experiences, especially for mothers who had more than one birth in the previous 5-year period. Additionally, mothers who did not remember the PNC care they received for either herself or her newborn or both were excluded from analysis. Finally, a total of 2405 mothers (between 15 and 49 years) were included in this study (**S1 Fig**).

### Outcome variable

To develop the outcome variable "respondent's health checked after discharge" and "baby postnatal checked within two months" were used as the starting point. Following this, "where respondent was checked after discharge" and "where the baby was first checked" were used to identify where the PNC was conducted after discharge. Those where either the mother or baby or both were checked in any health facility (public or private, and not necessarily the same at place of birth) were considered as returned for PNC use, while those who were not checked were considered as not returned for PNC use. Those who were checked (either the mother or baby or both) in home were excluded from the analysis. Finally, the outcome variable i.e. "Returned to health facility for PNC use" was developed with a value of "1 = Yes" if either of the mother or baby or both returned at least once for PNC check at any health facility and "0 = No" if neither the mother nor baby were checked at any health facility within 42 days of post-delivery for the mother and two months for the baby. The reason why we include the PNC use for babies up to two months after birth was explained elsewhere [20].

### Independent variables

The explanatory variables for this study were grouped in two subgroups; 1) socio-demographic (age, marital status, educational status, place of residence, region type, religion, sex of the household head, family size, age of the household head and media exposure) and socio-economic characteristics (wealth status and occupation), and 2) Gynecological/Obstetrical characteristics and service utilization variables (age at first sex, age at first birth, number of ever born children, pregnancy wantedness, number of ANC visits, delivery by caesarean section, checked before discharge, attitude towards domestic violence and informed when to return). Women's

age was grouped in to three categories: 15–24, 25–34 and ≥35 years. Region type was grouped in to three categories: Metropolitan for Addis-Ababa, Harrar and Drie-Dawa, Large central for Amhara, Orromia, South Nations and nationalities and Tigray, and Small peripheral for Afar, Benishangule, Gambella and Somalia. Residence was categorized as rural or urban. Marital status was grouped into two categories: Others (single, divorced and widowed) and married/living with husband. The highest level of education achieved by women was categorized in to four groups: no education, primary, secondary and higher. Religion of the participant was grouped in to four categories: Orthodox, Muslim, Protestant and others (for catholic and traditional). No categorization was done for wealth status; it was taken as per the EDHS data (poorest, poorer, middle richer and richest). Ownership of place of delivery was categorized in to two groups: governmental (for governmental hospital, governmental health center, governmental health post and other public sector) and non-governmental (for private hospital, private clinic, NGO health facility, other private and NGO). Family size was grouped into three categories: 1 to 4, 5 to 8 and ≥ 9 members. Age of the household head was grouped in to three categories: 16–29, 30–59 and 60–88 years. Regarding media exposure, we were grouped it in to four categories: not at all, less than once per week, at least once per week and almost every day. Furthermore, for occupation of the women, we took the respondent's grouped occupation and was categorized in to two groups: employed (for other than not working) and not employed (for not working). Age at first sex and age at first birth were categorized in to three groups: 8–14, 15–17 and ≥ 18 and 12–19, 20–24 and ≥25 years respectively. Number of ANC visits during pregnancy was categorized in to four groups: no visit, 1–3 visits, 4–7 visits and ≥ 8 visits. Furthermore, facility type for delivery was categorized in to three groups: health post/clinic/NGO health facility, health center and hospital. The category for autonomy (low and high) and attitude towards domestic violence (supporting and opposing) were explained elsewhere [22, 23].

## Data analysis

After categorizing and recoding of different variables, frequencies and proportions were reported to describe categorical variables using cross tabulation tables. Furthermore, texts and graphs were used to present the finding. To compensate the unequal probability of selection between the strata due to non-proportional allocation of samples to different regions, place of residence and non-response rate among participants, a weighted sample was used [14]. Since DHS data are hierarchical, i.e. individuals (level 1) were nested within communities (level 2); a two-level mixed-effect logistic regression model was fitted to estimate both independent (fixed) effect of the explanatory variables and community-level (random) effect on return for PNC utilization among mothers who gave birth at health facilities. The log of the probability of PNC utilization was modeled using a two-level multilevel model as follows(as indicated elsewhere) [22]:

$$\text{Log} \frac{\Pi ij}{[1 - \Pi ij]} = \beta 0 + \beta 1 Xij + \beta 2 Zij + \mu j + eij$$

Where, i and j are the level 1 (individual) and level 2 (community) units, respectively; X and Z refer to individual and community-level variables, respectively; πij is the probability of return to health facility for PNC utilization for the $i^{th}$ women in the $j^{th}$ community; the β's were the fixed coefficients. Whereas, β0 is the intercept-the effect on the probability of returning to health facility for PNC use in the absence of influence of predictors; and uj showed the random effect (effect of the community on returning to health facility for PNC use after health facility delivery) for the jth community and eij showed random errors at the individual

levels. Due to clustered data nature, the within and between community variations were taken in to account by assuming each community had different intercept (β0) and fixed coefficient (β).

During the advanced analysis, first we conduct unadjusted multilevel logistic regression analysis to identify selected variables for the next models. Second, we estimate the null-model (model-0) which only indicates the random intercept and allowed detecting the existence of a possible contextual dimension for returning to health facilities for PNC after health facility delivery [24]. Then, we include the individual and community level factors (with p-value of ≤0.25 in the unadjusted multilevel logistic regression) to develop models 1 and 2 respectively. Finally, individual and community level factors from model 1 and 2 were fitted (model-3) together to adjust the estimates of the separated models (models 1 and 2).

The measures of association (fixed-effects) estimate between the odds of women to return to health facility for PNC and other independent variables were reported using Adjusted Odds Ratio (AOR) with its 95% Confidence Interval (CI) and p-value of $\leq 0.05$ to declare the significance of the estimates. Furthermore, the measures of variation (random-effects) were reported using Intraclass Correlation Coefficient (ICC) to explain how much the observation in the same cluster were resembled each other [22], Median Odds Ratio (MOR) to measure of unexplained cluster heterogeneity [22] and Proportion of Change in Variance (PCV) to estimate the reduction in variance due to the step-wise introduction of variables into the model [25]. Moreover, Akaike Information Criterion (AIC) and over all percentage of correct classification were also reported.

To calculate, ICC, MOR and PCV, we used the following formulas as illustrated elsewhere [24]:

$$ICC = \frac{\delta 2}{\delta 2 + \frac{\pi 2}{3}} = \frac{\delta 2}{\delta 2 + 3.29}$$

Where $\delta 2$ is the area level variance and $\frac{\pi 2}{3}$ corresponds to individual level variance.

$$MOR = \exp(\sqrt{2 * \delta 2} * 0.6745 \simeq \exp(0.95\sqrt{\delta 2})$$

Where $\delta 2$ the area level variance and 0.6745 is the 75th centile of the cumulative distribution function of the normal distribution with mean 0 and variance 1.

$$PCV = \frac{\delta 2A - \delta 2B}{\delta 2A}$$

Where $\delta 2A$ = variance of the initial model, and $\delta 2B$ = variance of the model with more terms.

Multi-collinearity was checked using the Variance Inflation Factor (VIF) test and all variables were with value of <5 which indicates there was no multi-collinear variables in the model [26]. All the analysis was done using IBM-SPSS statistics version 21.

## Ethical consideration

Authorization to use the data was obtained from MEASURE DHS by providing a brief description of the study through their website (https://dhsprogram.com/data/). Approval for EDHS data utilization for this study was obtained from the data originator, ICF Macro International U.S.A. before the data was extracted from their web platform.

# Results

## Socio-demographic and socio-economic characteristics

Only 13% (n = 94) of youths who delivered at health facilities returned to health facilities for PNC. Above 27% (n = 61) of women from metropolitan regions returned to health facilities for PNC while 87% (n = 1814) of women from large central regions did not return. Furthermore, slightly higher proportions (22%) of women who delivered at non-governmental health facilities returned for PNC than women who delivered at governmental health facilities (14%). Additionally, 90% (n = 181) of women from family size of nine and above did not return for PNC while 17% (n = 167) of women from one to four family size returned for PNC after they gave birth at health facilities (**Table 1**).

## Gynecological/obstetrical characteristics

Above one third (36%; n = 49) of participants with eight and above ANC visits returned to health facilities for PNC after they gave birth at health facilities. On the other hand, above 90% (91.6%; n = 214) of participants who had not had ANC and delivered at health facilities did not return to health facilities for PNC. Furthermore, 19% (n = 125) of participants who delivered at hospitals returned to health facilities for PNC while 87% (n = 1320) of women who delivered at health centers did not return. One fourth (25%; n = 143) of the participants who were informed when to return returned for PNC to health facilities. Around one-third (31.7; n = 58) of the participants who delivered by caesarean section were returned to health facilities for PNC use (**Table 2**).

## Return to health facility for PNC

In this analysis, from the total 2405 participants, 14.3% ((95%CI: 12.1–16.8), (n = 344)) of them returned to health facilities for PNC use after they gave birth at a health facility and only 2.5% ((95%CI: 1.6–3.8), (n = 59)) women returned for PNC use for both the mother and the child (**S2 Fig**).

## Associated factors with return to health facility for PNC (fixed effects)

After adjusting for individual and community level factors in the final model (model 3), occupation and number of ANC visits from individual-level and caesarean section delivery and place of residence from community-level were the identified significant variables with returning to health facilities for PNC use among women who delivered at health facilities.

Employed women were 51% (AOR = 1.51, 95%CI: 1.04–2.19) more likely to return to health facility for PNC than not employed women during their postnatal period after they gave birth at health facilities. Furthermore, the odds of returning to the health facility for PNC after delivering at the health facility was 2.9 times (AOR = 2.90, 95%CI: 1.05–8.00) higher among women who had eight and above ANC visits than women who had no ANC visits during their pregnancy. Women who delivered by caesarean section were also in higher odds (AOR = 2.53, 95%CI: 1.40–4.58) of returning to health facilities for PNC than their counterparts during their postnatal period. On the other hand, rural resident women were in lower odds (AOR = 0.56, 95%CI: 0.36–0.88) of returning to health facility for PNC use than urban resident women (**Table 3**).

## Random-effect estimates

Two level mixed-effect logistic regression model was used to analyze the effect of individual and community level factors on returning to health facilities for PNC among women delivered

**Table 1. Socio-demographic and socio-economic characteristics of study participants, analysis from the 2016 EDHS, (N = 2405).**

| Variable | Returned for PNC | | Total N (%) | Unweight number |
|---|---|---|---|---|
| | No n (%) | Yes n (%) | | |
| **Age** | | | | |
| 15–24 | 615 (86.7) | 94 (13.3) | 709(100) | 794 |
| 25–34 | 1047(85.5) | 178(14.5) | 1225(100) | 1354 |
| > = 35 | 398(84.6) | 72(15.4) | 470(100) | 548 |
| **Region Type** | | | | |
| Metropolitan | 160(72.5) | 61(27.5) | 221(100) | 848 |
| Large central | 1814(87.1) | 268(12.9) | 2082(100) | 1229 |
| Small peripheral | 86(84.6) | 16(15.4) | 102(100) | 619 |
| **Place of residence** | | | | |
| Urban | 643(79.1) | 170(20.9) | 813(100) | 1255 |
| Rural | 1417(89.1) | 174(10.9) | 1591(100) | 1441 |
| **Marital status** | | | | |
| Other (single, widowed, divorced | 149(83.2) | 30(16.8) | 179(100) | 251 |
| Married/Living with partner) | 1911(85.9) | 314(14.1) | 2225(100) | 2445 |
| **Highest education level** | | | | |
| No education | 848(88.4) | 111(11.6) | 959(100) | 965 |
| Primary | 772(86.9) | 117(13.1) | 889(100) | 976 |
| Secondary | 282(82.5) | 60(17.5) | 342(100) | 462 |
| Higher | 157(73.6) | 56(26.4) | 213(100) | 293 |
| **Religion** | | | | |
| Orthodox | 954(81.1) | 223(18.9) | 1177(100) | 1234 |
| Muslim | 656(89.2) | 79(10.8) | 735(100) | 1009 |
| Protestant | 427(91.2) | 41(8.8) | 468(100) | 421 |
| Others | 24(97.1) | 1(2.9) | 25(100) | 32 |
| **Sex of household head** | | | | |
| Male | 1715(86.8) | 260(13.2) | 1975(100) | 2035 |
| Female | 346(80.4) | 84(19.6) | 430(100) | 661 |
| **Wealth index** | | | | |
| Poorest | 194(89.2) | 24(10.8) | 218(100) | 347 |
| Poorer | 349(90.5) | 37(9.5) | 386(100) | 350 |
| Middle | 364(87.3) | 53(12.7) | 417(100) | 324 |
| Richer | 407(88.4) | 53(11.6) | 460(100) | 351 |
| Richest | 747(80.7) | 178(19.3) | 925(100) | 1324 |
| **Ownership of place of delivery** | | | | |
| Non-governmental | 90(78.0) | 25(22.0) | 115(100) | 254 |
| Governmental | 1971(86.1) | 319(13.9) | 2290(100) | 2442 |
| **Family size** | | | | |
| 1 to 4 | 820(83.1) | 167(16.9) | 987(100) | 1142 |
| 5 to 8 | 1060(87.1) | 157(12.9) | 1217(100) | 1307 |
| 9 and above | 181(90.2) | 20(9.8) | 200(100) | 247 |
| **Age of the HH head** | | | | |
| 16–29 years | 516(85.5) | 87(14.5) | 603(100) | 700 |
| 30–59 years | 1366(85.6) | 231(14.4) | 1596(100) | 1767 |
| 60–88 years | 178(87.1) | 26(12.9) | 204(100) | 227 |
| **Media exposure** | | | | |
| Not at all | 959(87.3) | 139(12.7) | 1098(100) | 1108 |

*(Continued)*

**Table 1.** (Continued)

| Variable | Returned for PNC | | Total N (%) | Unweight number |
|---|---|---|---|---|
| | **No n (%)** | **Yes n (%)** | | |
| Less than once per week | 205(87.9) | 29(12.1) | 234(100) | 270 |
| At least once per week | 394(87.2) | 58(12.8) | 452(100) | 543 |
| Almost every day | 503(80.9) | 119(19.1) | 622(100) | 775 |
| **Occupation** | | | | |
| Not employed | 1041(88.5) | 135(11.5) | 1176(100) | 1372 |
| Employed | 1020(83.0) | 209(17.0) | 1229(100) | 1324 |

at health facilities. From the empty (null) model of **Table 4**, 32.6% of the variation in the odds of returning to health facilities for PNC use among women delivered at health facilities was due to cluster variation and this variability was declined to 30.8% in the final model. Thus, to explain the factors associated with the return to health facility for PNC, the final model was taken.

## Discussion

This study was aimed to determine the utilization of facility-based PNC and associated factors after discharged from a health facility among Ethiopian mothers who delivered at a health facility.

The finding showed that the overall utilization of facility-based PNC is 14.3% among women who delivered at a health facility. Being employed, greater number of ANC visits, caesarean section delivery, and rural residence were identified as factors associated with the use of facility-based PNC services in Ethiopia.

The magnitude of facility-based PNC in this study is lower than that of studies in some other low resource countries, which documented PNC coverage of 43% in Nepal and 50.9% in Malawi [17, 27], almost similar to results from studies conducted in Tanzania (18.1%) and Benin (18.4%) [12, 28], and slightly higher than finding from Tanzania (10.4%) and Rwanda (12.8%) [13, 29]. These differences could be due to differences in socio-demographic and socio-economic status of countries. Furthermore, there could be differences in the various studies reporting periods after delivery and different methods among the studies and setups. In this regard, our finding on 'return for facility-based PNC after discharge following health facility-based delivery' could be a more robust indicator for PNC service utilization and continuity of care, as it gives a picture of the health care seeking behavior of mothers after health facility delivery.

Regarding factors associated with facility-based PNC utilization, mothers who received ≥8 ANC visits during their pregnancy had a higher likelihood of utilizing facility-based PNC than those who did not have an ANC visit. Repeated ANC visits may instill greater sense of value in mothers regarding the potential benefit of contact with a provider, thus improving their health seeking behavior following delivery [19, 27]. Furthermore, repeated contact with health workers during pregnancy through ANC services could promote confidence and familiarity with the health system leading to increased trust in the health system [12, 30, 31]. On the other hand, those mothers with more ANC visit (≥8 visits) could also have more complications than their counterparts [1, 9, 12, 13]. This finding is also a great opportunity to support the new WHO recommendation of increasing the frequency of focused ANC to eight visits [32].

Another variable found to be positively associated with facility-based PNC was caesarean section delivery. Those who gave birth through caesarean section were 2.53 times more likely

**Table 2. Gynecological/Obstetrical characteristics and related service utilization among participants, analysis from the 2016 EDHS, (N = 2405).**

| Variable | Returned for PNC | | Total N (%) | Unweight number |
|---|---|---|---|---|
| | No n (%) | Yes n (%) | | |
| **Number of ever born children** | | | | |
| 1–4 | 1568(84.9) | 279(15.1) | 1847(100) | 2095 |
| 5–8 | 410(87.8) | 57(12.2) | 467(100) | 511 |
| > = 9 | 83(91.4) | 8(8.6) | 90(100) | 90 |
| **Age at first birth** | | | | |
| 12–19 years | 1142(87.4) | 164(12.6) | 1306(100) | 1402 |
| 20–24 years | 678(84.8) | 122(15.2) | 800(100) | 910 |
| > = 25 years | 240(80.5) | 58(19.5) | 299(100) | 384 |
| **Age at first sex** | | | | |
| 8–14 years | 319(86.0) | 52(14.0) | 371(100) | 425 |
| 15–17 years | 889(87.8) | 123(12.2) | 1012(100) | 1097 |
| > = 18 years | 853(83.5) | 169(16.5) | 1022(100) | 1174 |
| **Wanted pregnancy when became pregnant** | | | | |
| Then | 1540(85.1) | 271(14.9) | 1811(100) | 2132 |
| Later | 366(86.8) | 55(13.2) | 421(100) | 414 |
| No more | 155(89.4) | 18(10.6) | 173(100) | 150 |
| **Autonomy** | | | | |
| Low | 665(84.8) | 119(15.2) | 784(100) | 865 |
| High | 1395(86.1) | 226(13.9) | 1621(100) | 1831 |
| **Number of ANC** | | | | |
| No ANC | 214(91.6) | 20(8.4) | 234(100) | 180 |
| 1–3 ANC | 719(89.5) | 85(10.5) | 804(100) | 820 |
| 4–7 ANC | 1039(84.5) | 191(15.5) | 1230(100) | 1469 |
| > = 8 ANC | 88(64.1) | 49(35.9) | 137(100) | 227 |
| **Facility type for delivery** | | | | |
| HP/Clinic/NGO HF | 216(90.6) | 23(9.4) | 239(100) | 243 |
| Health center | 1320(87.0) | 197(13.0) | 1517(100) | 1402 |
| Hospital | 525(80.8) | 125(19.2) | 649(100) | 1051 |
| **Delivered by caesarean section** | | | | |
| No | 1936(87.1) | 286(12.9) | 2222(100) | 2441 |
| Yes | 125(68.3) | 58(31.7) | 183(100) | 255 |
| **Child or/and mother checked before discharge after delivery** | | | | |
| No | 1017(92.0) | 88(8.0) | 1105(100) | 1073 |
| Yes | 1000(79.8) | 253(20.2) | 1253(100) | 1582 |
| Not remembered | 44(92.9) | 3(7.1) | 47(100) | 41 |
| **Attitude towards domestic violence** | | | | |
| Supporting domestic violence | 1225(87.7) | 172(12.3) | 1397(100) | 1383 |
| Opposing domestic violence | 835(82.9) | 173(17.1) | 1008(100) | 1313 |
| **Informed when to return** | | | | |
| No | 269(85.1) | 47(14.9) | 316(100) | 334 |
| Yes | 429(75.0) | 143(25.0) | 572(100) | 788 |
| Do not remember | 1363(89.8) | 154(10.2) | 1517(100) | 1574 |

to return for health facility-based PNC after discharge. This finding is in line with the findings of other studies [9, 11, 12]. Caesarean section delivery could affect the healthcare seeking of mothers as part of recommended follow-up service [7, 12] and due to increased risk of complications [11, 33].

**Table 3. Multilevel logistic regression analysis for factors associated with returning to health facility for postnatal care utilization among mothers who delivered in health facility in Ethiopia: Analysis of the 2016 EDHS.** Individual and community level characteristics.

| | COR (95%CI) | Model 0 | Model 1 | Model 2 | Model 3 |
|---|---|---|---|---|---|
| **Highest education level** | | | | | |
| No education | 1 | | 1 | | 1 |
| Primary | 1.09(0.68–1.73) | | 1.05(0.65–1.70) | | 1.00(0.62–1.63) |
| Secondary | 1.36(0.77–2.39) | | 1.14(0.61–2.13) | | 1.17(0.61–2.31) |
| Higher | 2.47(1.45–4.19) | | 1.61(0.82–3.13) | | 1.30(0.59–2.86) |
| **Religion** | | | | | |
| Orthodox | 1 | | 1 | | 1 |
| Muslim | 0.65(0.41–1.02) | | 0.83(0.51–1.38) | | 0.84(0.49–1.43) |
| Protestant | 0.66(0.38–1.14) | | 0.72(0.42–1.22) | | 0.67(0.38–1.17) |
| Others | 0.23(0.02–2.22) | | 0.26(0.03–2.57) | | 0.32(0.03–3.55) |
| **Age at first birth** | | | | | |
| 12–19 years | 1 | | 1 | | 1 |
| 20–24 years | 1.40(0.96–2.04) | | 1.09(0.68–1.73) | | 1.35(0.89–2.05) |
| > = 25 years | 1.63(0.95–2.78) | | 1.07(0.59–1.93) | | 1.18(0.66–2.11) |
| **Sex of household head** | | | | | |
| Male | 1 | | 1 | | 1 |
| Female | 1.31(0.87–1.97) | | 1.18(0.80–1.75) | | 1.05(0.71–1.56) |
| **Wealth index** | | | | | |
| Poorest | 1 | | 1 | | 1 |
| Poorer | 1.05(0.54–2.05) | | 1.17(0.60–2.29) | | 1.33(0.70–2.53) |
| Middle | 1.04(0.52–2.08) | | 1.18(0.58–2.38) | | 1.32(0.64–2.74) |
| Richer | 0.86(0.42–1.75) | | 0.91(0.45–1.85) | | 1.02(0.50–2.09) |
| Richest | 1.60(0.80–3.21) | | 1.35(0.61–2.97) | | 0.80(0.28–2.30) |
| **Occupation** | | | | | |
| Not employed | 1 | | 1 | | 1 |
| Employed | 1.52(1.06–2.18) | | 1.32(0.90–1.91) | | **1.51(1.04–2.19)**[*] |
| **Number of ANC** | | | | | |
| No ANC | 1 | | 1 | | 1 |
| 1–3 ANC | 1.24(0.47–3.25) | | 1.15(0.44–3.05) | | 1.42(0.55–3.65) |
| 4–7 ANC | 1.64(0.66–4.09) | | 1.42(0.55–3.65) | | 1.54(0.62–3.84) |
| > = 8 ANC | 4.04(1.52–10.76) | | **3.11(1.11–8.70)**[*] | | **2.90(1.05–8.00)**[*] |
| **Age at first sex** | | | | | |
| 8–14 years | 1 | | 1 | | 1 |
| 15–17 years | 091(0.53–1.56) | | 0.91(0.52–1.59) | | 0.93(0.52–1.67) |
| > = 18 years | 1.46(0.87–2.46) | | 1.23(0.66–2.32) | | 1.21(0.65–2.26) |
| **Media exposure** | | | | | |
| Not at all | 1 | | 1 | | 1 |
| Less than once per week | 0.79(0.43–1.47) | | 0.73(0.38–1.40) | | 0.60(0.29–1.23) |
| At least once per week | 0.84(0.48–1.49) | | 0.69(0.36–1.31) | | 0.65(0.37–1.15) |
| Almost every day | 1.34(0.82–2.21) | | 0.88(0.44–1.68) | | 0.77(0.42–1.41) |
| **Region Type** | | | | | |
| Metropolitan | 1 | | | 1 | 1 |
| Large central | 0.28(0.17–0.45) | | | 0.58(0.31–1.08) | 0.72(0.36–1.14) |
| Small peripheral | 0.61(0.18–2.09) | | | 1.07(0.27–4.28) | 1.32(0.31–5.62) |
| **Place of residence** | | | | | |
| Urban | 1 | | | 1 | 1 |

*(Continued)*

**Table 3.** (Continued)

|  | COR (95%CI) | Model 0 | Model 1 | Model 2 | Model 3 |
|---|---|---|---|---|---|
| Rural | 0.36(0.24–0.55) |  |  | **0.56(0.33–0.94)*** | **0.56(0.36–0.88)*** |
| **Facility type for delivery** |  |  |  |  |  |
| HP/Clinic/NGO HF | 1 |  |  | 1 | 1 |
| Health center | 1.48(0.62–3.51) |  |  | 1.45(0.62–3.39) | 1.45(0.61–3.45) |
| Hospital | 2.07(0.79–5.39) |  |  | 1.21(0.47–3.15) | 1.23(0.46–3.27) |
| **Delivered by caesarean section** |  |  |  |  |  |
| No | 1 |  |  | 1 |  |
| Yes | 3.43(1.87–6.32) |  |  | **2.79(1.46–5.35)*** | **2.53(1.40–4.58)** |
| **Child or/and mother checked before discharge after delivery** |  |  |  |  |  |
| No | 1 |  |  | 1 | 1 |
| Yes | 2.47(1.54–3.94) |  |  | 1.55(0.89–2.73) | 1.62(0.92–2.85) |
| Not remembered | 0.70(0.17–2.87) |  |  | 0.63(0.15–2.75) | 0.57(0.13–2.41) |
| **Informed when to return** |  |  |  |  |  |
| No | 1 |  |  | 1 | 1 |
| Yes | 1.82(1.03–3.24) |  |  | 1.74(0.99–3.08) | 1.75(0.99–3.10) |
| Do not remember | 0.67(0.37–1.18) |  |  | 0.97(0.53–1.78) | 1.03(0.56–1.86) |

* $P<0.05$

** $P<0.005$.

Occupation was another positively associated variable; employed participants were in higher odds of receiving health facility based PNC after they discharged for their health facility based delivery. This finding is in line with the finding of other study [31]. Employed women could have better opportunity to pay the transport cost, and they could have better educational level for which they were able to be employed.

Finally, rural residence was negatively associated with return to health facilities for PNC among women who gave birth at health facilities. Those who lived in rural were 44% less likely to return to health facilities for PNC use than their counterparts. This finding is also in line with the finding of other studies [17]. This could be explained by different reasons like: long distance between home and health facility [34], transport inaccessibility, inability to cover transport cost [16, 31], perception of good health [16, 35], poor autonomy among women to decide on service use and other cultural influences [16, 36].

Our study made use of cross-sectional data from the 2016 Ethiopian Demographic and Health Survey. The data relies on women's self-reported care utilization, and may be influenced by recall bias, given that the study events took place within the 5 years preceding the survey. However, the study has a number of strengths. The data is national survey data, and the

**Table 4. Measure of variation on individual and community level factors among health facility delivered mothers in Ethiopia, EDHS 2016 dataset.**

| Measures | Model 0 | Model 1 | Model 2 | Model 3 |
|---|---|---|---|---|
| **AIC**(Akaike information criterion) | 12159.59 | 12360.61 | 12442.97 | 12491.91 |
| **Over all percentage of correct classification** | 87.0% | 87.3% | 88.2% | 88.0% |
| **Variance** | 1.594 | 1.472 | 1.473 | 1.469 |
| **VPC or ICC** (Variance partition coefficient/ Intraclass Correlation Coefficient) | 0.326 | 0.309 | 0.309 | 0.308 |
| **PCV (Proportion of change in variance) (%)** | Ref | 7.65 | 7.59 | 7.84 |
| **MOR** (Median Odds Ratio) | 3.33 | 3.18 | 3.18 | 3.17 |

sample size is powered to be generalizable at national and regional level. Furthermore, the method of analysis was multilevel which adjust for individual and community level effects. Thus, this study could give more robust information about the health seeking choice of mothers on facility-based PNC use and help to design proper strategy to boost PNC service utilization.

## Conclusion

The finding of this study showed that facility-based PNC utilization among mothers who delivered at health facilities is still low in Ethiopia. Having $\geq$ 8 ANC visits, caesarean section delivery, being employed and rural residence were the identified factors associated with facility-based PNC service utilization. Both individual and community level variables determined women to return to health facilities for PNC use. Therefore, adopting context-specific strategies/policies could improve PNC utilization and should be paid a due focus.

## Supporting information

**S1 Fig. Schematic presentation of sampling procedure.**
(TIF)

**S2 Fig. Magnitude of women who returned to health facility for PNC use after they gave birth at health facility.**
(TIF)

## Acknowledgments

We would like to thank Central Statistical Agency (CSA) and MEASURE DHS project for providing free access to the data.

## Author Contributions

**Conceptualization:** Brhane Ayele, Mulugeta Woldu, Haftom Gebrehiwot, Tsegay Wellay.

**Data curation:** Brhane Ayele.

**Formal analysis:** Brhane Ayele, Tsegay Wellay.

**Investigation:** Brhane Ayele.

**Methodology:** Brhane Ayele, Mulugeta Woldu, Haftom Gebrehiwot.

**Project administration:** Brhane Ayele.

**Resources:** Brhane Ayele.

**Software:** Brhane Ayele.

**Validation:** Brhane Ayele.

**Visualization:** Sarah Hurlburt.

**Writing – original draft:** Brhane Ayele.

**Writing – review & editing:** Brhane Ayele, Mulugeta Woldu, Haftom Gebrehiwot, Tsegay Wellay, Tsegay Hadgu, Hailay Gebretnsae, Alemnesh Abrha, Equbay Gebre-egziabher, Sarah Hurlburt.

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
