## [Decision Letter · Decision Letter 0]

8 Sep 2020

PONE-D-20-18442

Postnatal care follow-up among mothers who delivered at health facilities in Ethiopia: Further analysis of the 2016 Demographic and Health Survey.

PLOS ONE

Dear Dr. Ayele,

Thank you for submitting your manuscript to PLOS ONE. After careful consideration, we feel that it has merit but does not fully meet PLOS ONE’s publication criteria as it currently stands. Therefore, we invite you to submit a revised version of the manuscript that addresses the points raised during the review process.

At least three published studies (Sisay et al, 2018; Ayele et al 2019; Fikadu et al, 2019) have so far identified predictors of PNC utilization in Ethiopia based on the same dataset (Ethiopian DHS 2016)bused in this study. What was the justification for reanalyzing the dataset again?The operational definition used for postnatal care (any health check-up in the first 42 days) is deficient because it does not takethe timing and number of PNC visits into consideration. At least basic description on the timing and frequency of PNC visits must be given in the manuscript.The authors provided inadequate information how they analyzed the data. As the DHS employed cluster sampling design, data must be analyzed using analytical techniques (e.g. mixed effects model, GEE or survey design) that accommodate the clustering nature of the study. Furthermore, sampling weights provided in the data and post-stratification weight for balancing the contribution of the regions must be used.The reason for including some of the variables included in the multivariable is not clear. The variables “Children born in last five years” and “Children born in last one year” are likely to have redundant information in the model and hence one of them has to be dropped. Further, while analyzing these variables, the authors should rather be focused on number of births that occur before the index pregnancy analysed for the PNC. Births that occur after the index birth should be excluded and this can easily be done in DHS by taking date of birth into consideration.The variable “Currently working” is also confusing. Does it refer to the occupation of the mother during the index pregnancy? How can the current occupation affect PNC received some years back?Two variables (ownership of mobile phone and having bank account) are already used for developing the wealth index. So, what is the purpose of considering them as distinct variables again?It is not clear how 50% of the data fall into the richest wealth quintile (Table 1). As I understand, wealth index in DHS data divides the study subjects into 5 equal Quintiles each nearly having 20% of relative frequency. I recommend the authors to redo the PCA again.Fitting 18 variables in one model would be to much and might result in an unstable model. I recommend the authors to remove less relevant variables or to use more stringent p-value for screening candidate variables for the multivariable model.

We look forward to receiving your revised manuscript.

Kind regards,

Samson Gebremedhin, PhD

Academic Editor

PLOS ONE

Journal Requirements:

2. Thank you for submitting the above manuscript to PLOS ONE. During our internal evaluation of the manuscript, we found significant text overlap between your submission (mainly methods and 'limitations' sections) and the following previously published works, some of which you are an author.

https://reproductive-health-journal.biomedcentral.com/articles/10.1186/s12978-019-0818-2

Please revise the manuscript to rephrase the duplicated text, cite your sources, and provide details as to how the current manuscript advances on previous work. Please note that further consideration is dependent on the submission of a manuscript that addresses these concerns about the overlap in text with published work.

We will carefully review your manuscript upon resubmission, so please ensure that your revision is thorough

Additional Editor Comments (Section-by-section comments):

Abstract

The operational definition for “utilization of PNC” should be providedThe conclusion sub-section should be condensed and should not repeat what has been reported in the results.

Background

Please don’t direct copy from other literature. Many sentences and paragraphs have not been adequately paraphrased.

Methods

Please describe the basic sampling technique employed in the DHS survey.Line 92-93: “Only the most recent child of the women was included in the analysis”. Please clearly indicate that this is about the most recent birth that happened in the preceding 5 years of the survey.Figure 1: I think the decision to exclude women who had home PNC is wrong. The data of these subjects should be retained in the analysis but they should be considered as they have no PNC.

Results

Socio-demographic and socio-economic characteristics: Looks very superficial. Please expandPlease interpret OR>1 on multiplicative, rather than additive scale.Table 3: “pace of delivery” >> “place of delivery”

Discussion

Please integrate the Strength and Limitation section with the Discussion section.Paragraph 2: is just a repetition of the findings of the study, I don’t see any discussionParagraph 3: Comparison should only be made with studies that estimated PNC coverage among women who had health facility delivery.

Reviewers' comments:

Reviewer's Responses to Questions

**Comments to the Author**

1. Is the manuscript technically sound, and do the data support the conclusions?

Reviewer #1: Yes

Reviewer #2: Yes

2. Has the statistical analysis been performed appropriately and rigorously? 

Reviewer #1: No

Reviewer #2: Yes

3. Have the authors made all data underlying the findings in their manuscript fully available?

Reviewer #1: Yes

Reviewer #2: Yes

4. Is the manuscript presented in an intelligible fashion and written in standard English?

Reviewer #1: Yes

Reviewer #2: No

5. Review Comments to the Author

Reviewer #1: Thank you for this well written manuscript. Just a few minor comments and questions:

Outcome variable

• Line 108: Please clarify whether the PNC being discussed is for the mother, the child or both

• It is not quite clear to me what is meant by “follow up check”. Is the paper focussing on immediate postnatal care or any postnatal care. Follow up would essentially mean that the mother got a first check up, and then they had a follow up check after a few days/weeks? Is this the case, or are the investigators only considering the first check? This needs to be clarified both when discussing the derivation of the outcome variable but might also need to be reflected in the topic

• At some point the authors have mentioned that women who were checked within a month were considered to have accessed PNC. However, the WHO guidelines stipulate that PNC should be accessed in the first 42 days after birth. Can the authors please clarify this, or correct this in their generation of the outcome variable?

Statistical analysis

• Demographic and Health Surveys are usually sampled at two levels. The authors have not mentioned that they adjusted their analysis for sample weights. How did the authors account for the multilevel structure of the data?

• Line 91: DHS does not have a specific data set for pregnancy and postnatal care. Can the authors clarify in the write up whether they used the birth recode or the women’s recode? This will be useful for reproducibility of the results by the readers. The DHS does not have a unique file for pregnancy and postnatal care

• Line 115 and 116: I would expect the following variables to be highly collinear: (number of ever born children, children born in last five years, children born in last one 116 year). Is there a particular reason why you chose to include all three? What was the VIF for these variables?

• What bivariate method was used? This should be specified in the write up. I have not seen results of any bivariate analysis presented. What the authors have presented is a univariate logistic regression analysis

• In the methods section there is a mention of the VIF, but I don’t see the results from the VIF being talked about anywhere. Were any of the variables highly correlated or not? A sentence on this might be useful when starting to discuss results from the logistic regression

Table 1

Ownership of pace of delivery should have place of delivery instead of pace

Table 2

• Other(single, widowed, divorce): divorce should be divorced to be consistent with the other words

Table 3

• Part 3 of the caption of Table 3 should read as a table that is continuing from the previous pages. Just as it was done on the second part of the table

• P-values are usually categorized as follows: *** p-value <0.001, ** p-value < 0.01, * p-value < 0.005. The authors might wish to adopt this

• On the same p-value note,

• Why is it that the statistically significant results in the adjusted logistic regression model have been highlighted but those in the unadjusted model have not?

• I find Table 3 to be a little crowded. Firstly, the table caption says that the table presents results from the logistic regression, and yet the table includes results from cross-tabulations

• Normally, the cross tabulations would be included in Table 1. i.e. one could have 3 columns in the table: had PNC (n & %), did not have PNC (n & %), total (which is what is currently the column in Table 1 (n & %))

Reviewer #2: 1. The manuscript is technically sound because it covers all the aspects of a good manuscript and content stuck to the subject matter. The data supports the conclusion methodology, results (proportions and AOR) appropriately handled and presented to draw the conclusion.

2. Statistical analysis well handled - explanations of process and methods appropriately selected and handled.

3. The Author clearly indicated that the data will be made available without restrictions.

4. Lines 69 and 70, the sentence is not clear, "not advised to return to more complications among mothers". Line 81 consideration of removing the word despite and make the point more clear. Line 151 to rephrase the sentence and add - women who were not checked before discharge.

6. PLOS authors have the option to publish the peer review history of their article (what does this mean?). If published, this will include your full peer review and any attached files.

Reviewer #1: No

Reviewer #2: **Yes: **Dr. Charles Chungu

---

## [Author Response · Author response to Decision Letter 0]

2 Nov 2020

Dear editor and reviewers thank you very much for your comments and suggestions. Your concerns are very important and critical to improve our manuscript accordingly. We tried to address the points in our revised manuscript. Here are our point by point responses! 

Editor’s Comments :

At least three published studies (Sisay et al, 2018; Ayele et al 2019; Fikadu et al, 2019) have so far identified predictors of PNC utilization in Ethiopia based on the same dataset (Ethiopian DHS 2016)bused in this study. What was the justification for reanalyzing the dataset again?

Authors response :

Thank you for the comment and for raising the issue. Dear editor, though we make it clear in the current manuscript, the reason that we analyze the data is that our study population in our analysis and the study population in the mentioned studies are different. For example, the study population in Sisay et’al, 2018 was any women in childbearing age that gave birth in the last 5 years preceding the survey in the selected EAs irrespective of place of delivery. The study population in Ayele et’al, 2019 was also women in child bearing age that gave birth in the last 5 years preceding the survey and delivered at home. The study population in Fikadu et’al, 2019 was mothers who delivered at health facility and only those who had fourth antenatal care visit. Furthermore, there is also a recently published study (Tiruneh, 2020) on postnatal care utilization with a study population of women who give birth in both health facility and home. 

Thus, as our study is only included women who gave birth in health facilities it is different from all the above mentioned four articles. 

Editor’s Comments :

The operational definition used for postnatal care (any health check-up in the first 42 days) is deficient because it does not take the timing and number of PNC visits into consideration. At least basic description on the timing and frequency of PNC visits must be given in the manuscript.

Authors response :

If I understand your concern, you expect us to explain the timing and frequency of PNC visit. Thank you! However in this study the main objective that we want to explain is not the frequency and timing of PNC. We need to show how much mothers (those who delivered in health facilities) returned to any health facility (could be governmental, private, to the same facility where they gave birth or to other health facility) at least once before their 42 days of postpartum period. That’s why we did not include the frequency and timing of the visit. 

Editor’s Comments :

The authors provided inadequate information how they analyzed the data. As the DHS employed cluster sampling design, data must be analyzed using analytical techniques (e.g. mixed effects model, GEE or survey design) that accommodate the clustering nature of the study. Furthermore, sampling weights provided in the data and post stratification weight for balancing the contribution of the regions must be used.

Authors response :

Thank you very! 

The data analyzing method is changed according to your comment. Thus, we used mixed effect model to analyze the data so as to accommodate the clustering of the data. Furthermore, we also used sampling weigh to balance the regional variation. Thank you once again! 

Editor’s Comments :

The reason for including some of the variables included in the multivariable is not clear. The variables “Children born in last five years” and “Children born in last one year” are likely to have redundant information in the model and hence one of them has to be dropped. Further, while analyzing these variables, the authors should rather be focused on number of births that occur before the index pregnancy analyzed for the PNC. Births that occur after the index birth should be excluded and this can easily be done in DHS by taking date of birth into consideration.

Authors response :

Acceptable comment. The analysis already changed and this comment is taken it account. Thank you very much for your concern. 

Editor’s Comments :

The variable “Currently working” is also confusing. Does it refer to the occupation of the mother during the index pregnancy? How can the current occupation affect PNC received some years back?

Authors response :

Thank you! However, DHS data collects the socio demographic characteristics (like age, educational level, working status and wealth status) of the participants during the data collection. They can’t be collected in relation to specific event. That’s why we took the work status. However, to prevent confusion among readers we prefer the “Occupation” to “Currently working”. 

Editor’s Comments :

Two variables (ownership of mobile phone and having bank account) are already used for developing the wealth index. So, what is the purpose of considering them as distinct variables again?

Authors response :

Thank you! Acceptable and these variables are omitted in this manuscript. 

Editor’s Comments :

It is not clear how 50% of the data fall into the richest wealth quintile (Table 1). As I understand, wealth index in DHS data divides the study subjects into 5 equal Quintiles each nearly having 20% of relative frequency. I recommend the authors to redo the PCA again.

Authors response :

Method of analysis is changed and your concern is already solved. Thank you very much! 

Editor’s Comments :

Fitting 18 variables in one model would be too much and might result in an unstable model. I recommend the authors to remove less relevant variables or to use more stringent p-value for screening candidate variables for the multivariable model.

Authors response :

In the revised manuscript 15 variables are fitted in the mixed effect multilevel logistic regression model. Thank you for your valuable concern! 

Additional Editor’s Comments (Section-by-section comments):

Editor’s Comments :

Abstract 

The operational definition for “utilization of PNC” should be provided

The conclusion sub-section should be condensed and should not repeat what has been reported in the results.

Authors response :

Thank you!

The introduction and the conclusion of the manuscript are revised according to your suggestions. 

Editor’s Comments :

Background

Please don’t direct copy from other literature. Many sentences and paragraphs have not been adequately paraphrased.

Authors response :

Paraphrasing is conducted as much as possible. Thank you! 

Editor’s Comments :

Methods

Please describe the basic sampling technique employed in the DHS survey.

Authors response :

Acceptable comment and basic sampling technique of the DHS survey is included in the revised manuscript. Thank you! 

Editor’s Comments :

Line 92-93: “Only the most recent child of the women was included in the analysis”. Please clearly indicate that this is about the most recent birth that happened in the preceding 5 years of the survey.

Authors response :

Thank you! The comment is acceptable and included. 

Editor’s Comments :

Figure 1: I think the decision to exclude women who had home PNC is wrong. The data of these subjects should be retained in the analysis but they should be considered as they have no PNC.

Authors response :

Thank you for the comment!

However, women who received PNC at home were excluded due to the reason that the interest of this manuscript is mainly to show the health seeking of mothers for PNC after they gave birth in health facilities which could be decreased if they received it in home. In other words why they need to return to health facility for PNC reason if they received it in their home? They don’t! That’s why we exclude them from the analysis. We revised the figure to make it more clear in the revised manuscript. 

Editor’s Comments :

Results

Socio-demographic and socio-economic characteristics: Looks very superficial. Please expand

Please interpret OR>1 on multiplicative, rather than additive scale.

Table 3: “pace of delivery” >> “place of delivery”

Authors response :

Thank you very much! In the revised manuscript all of these comments are considered! Please see the revised version of the manuscript. 

Editor’s Comments :

Discussion

Please integrate the Strength and Limitation section with the Discussion section.

Paragraph 2: is just a repetition of the findings of the study, I don’t see any discussion

Paragraph 3: Comparison should only be made with studies that estimated PNC coverage among women who had health facility delivery.

Authors response :

Thank you for the comment. As you said in the strength and limitation part is integrated with the discussion part. 

The second paragraph is the repetition of the result. Yes, this part (the second paragraph) is the summary of the finding which is important to grasp the attention of the readers what the key findings of the study and their discussion points are followed subsequently. 

This is acceptable comment 

Reviewers comments 

Reviewer #1: Thank you for this well written manuscript. Just a few minor comments and questions:

Outcome variable

• Line 108: Please clarify whether the PNC being discussed is for the mother, the child or both

Authors response :

First I thank you very much for your precious comments!

The outcome variable which is “Return to health facility for PNC” in the revised manuscript is for both the mother and the baby. It ‘yes’ if the mother returned to use PNC either for her baby or herself or both. This is clearer in the revised version of the manuscript. 

Reviewers comments 

• It is not quite clear to me what is meant by “follow up check”. Is the paper focusing on immediate postnatal care or any postnatal care? Follow up would essentially mean that the mother got a first check up, and then they had a follow up check after a few days/weeks? Is this the case, or are the investigators only considering the first check? This needs to be

clarified both when discussing the derivation of the outcome variable but might also need to be reflected in the topic

Authors response :

Thank you! 

The paper’s focus is on PNC seeking among women who delivered in health facility; we do not focus on whether they received PNC before discharge or not. We only focus on ‘do mothers returned for PNC use to health facilities after they gave birth in health facilities at least once in their postpartum period?’ which can show the health seeking behavior of mothers for PNC. To make it more clear your comment is very important and we amend the title of the manuscript as well considering your suggestion. Thank you once again! 

Reviewers comments 

• At some point the authors have mentioned that women who were checked within a month were considered to have accessed PNC. However, the WHO guidelines stipulate that PNC should be accessed in the first 42 days after birth. Can the authors please clarify this, or correct this in their generation of the outcome variable?

Authors response :

If I understand your comment well, as you said the PNC should be accessed in the first 42 days after birth with a recommended frequency of four. Thus, in our manuscript we also took this period in to account regardless of the frequency; whether mothers were returned for PNC use after they gave birth at health facilities. Furthermore, the PNC for babies was considered up to two months after birth for explained reason. 

Reviewers comments 

Statistical analysis

• Demographic and Health Surveys are usually sampled at two levels. The authors have not mentioned that they adjusted their analysis for sample weights. How did the authors account for the multilevel structure of the data?

Authors response :

Thank you very much!

All your comments are acceptable and the revised version of the manuscript includes considerations to the mentioned (sampling weight and multilevel structure) issues. 

Reviewers comments

• Line 91: DHS does not have a specific data set for pregnancy and postnatal care. Can the authors clarify in the write up whether they used the birth recode or the women’s recode? This will be useful for reproducibility of the results by the readers. The DHS does not have a unique file for pregnancy and postnatal care

Authors response :

Thank you your comment is acceptable and included in the revised manuscript. The “Individual Record (IR) dataset was used! 

Reviewers comments 

• Line 115 and 116: I would expect the following variables to be highly collinear: (number of ever born children, children born in last five years, children born in last one 116 year). Is there a particular reason why you chose to include all three?

What was the VIF for these variables?

Authors response :

Thank you!

In the revised version your concerns are addressed! Unfortunately these variables are not included in the new version due to the reason that these variables were not fulfilling the assumption for inclusion. 

Reviewers comments 

• What bivariate method was used? This should be specified in the write up. I have not seen results of any bivariate analysis presented. What the authors have presented is a univariate logistic regression analysis. 

Authors response :

Thank you!

Bivariable multilevel logistic regression was used and clearly stated in the revised manuscript. 

Reviewers comments 

• In the methods section there is a mention of the VIF, but I don’t see the results from the VIF being talked about anywhere. Were any of the variables highly correlated or not? A sentence on this might be useful when starting to discuss results from the logistic regression

Authors response :

Thank you! 

VIF was done and no variables were above 5 and this is clearly stated in the revised document. 

Reviewers comments 

Table 1

Ownership of pace of delivery should have place of delivery instead of pace

Authors response :

Thank you!

Unfortunately this variable is omitted in the revised version as we only put variables which fulfill the assumption for inclusion. 

Reviewers comments 

Table 2

• Other(single, widowed, divorce): divorce should be divorced to be consistent with the other words

Authors response :

Corrected! 

Reviewers comments 

Table 3

• Part 3 of the caption of Table 3 should read as a table that is continuing from the previous pages. Just as it was done on

the second part of the table

Authors response :

Corrected! 

Reviewers comments 

• P-values are usually categorized as follows: *** p-value <0.001, ** p-value < 0.01, * p-value < 0.005. The authors might

wish to adopt this on the same p-value note,

Authors response :

Thank you for your suggestions! However, as the result of p-value in the final model rages from 0.002 to 0.04 for significant variables, we adapt P-value < 0.005** and 0.05* respectively. 

Reviewers comments 

• Why is it that the statistically significant results in the adjusted logistic regression model have been highlighted but those in the unadjusted model have not?

Authors response :

Corrected! 

Reviewers comments 

• I find Table 3 to be a little crowded. Firstly, the table caption says that the table presents results from the logistic

regression, and yet the table includes results from cross-tabulations

• Normally, the cross tabulations would be included in Table 1. i.e. one could have 3 columns in the table: had PNC (n & %), did not have PNC (n & %), total (which is what is currently the column in Table 1 (n & %))

Authors response :

Thank you very much! It is acceptable comment and corrected accordingly. Please see the revised version of the document. 

Reviewers comments 

Reviewer #2: 1. The manuscript is technically sound because it covers all the aspects of a good manuscript and content stuck to the subject matter. The data supports the conclusion methodology; results (proportions and AOR) appropriately handled and presented to draw the conclusion.

2. Statistical analysis well handled - explanations of process and methods appropriately selected and handled.

3. The Author clearly indicated that the data will be made available without restrictions.

4. Lines 69 and 70, the sentence is not clear, "not advised to return to more complications among mothers". Line 81 consideration of removing the word despite and make the point more clear. Line 151 to rephrase the sentence and add - women who were not checked before discharge.

Authors response :

Thank you very much!

The phrase "not advised to return to more complications among mothers" in line 69-70 is revised and we tried to make it more understandable. 

The “despite” in line 81 is amended and we tried to make it clear.

Accepted and your concern is considered.

Dear editor and reviewers, once again we thank you very much for all your valuable comments!

Kindly regards, 

Brhane Ayele (on behalf of all authors)

---

## [Decision Letter · Decision Letter 1]

11 Dec 2020

PONE-D-20-18442R1

Do mothers who delivered at health facilities return to health facilities for postnatal care follow-up? A multilevel analysis of the 2016 Ethiopian Demographic and Health Survey.

PLOS ONE

Dear Dr. Ayele,

Thank you for submitting your manuscript to PLOS ONE. After careful consideration, we feel that it has merit but does not fully meet PLOS ONE’s publication criteria as it currently stands. Therefore, we invite you to submit a revised version of the manuscript that addresses the points raised during the review process. Please also make sure that the manuscript is thoroughly edited for typographical and grammatical errors.  

We look forward to receiving your revised manuscript.

Kind regards,

Samson Gebremedhin, PhD

Academic Editor

PLOS ONE

Reviewers' comments:

Reviewer's Responses to Questions

**Comments to the Author**

1. If the authors have adequately addressed your comments raised in a previous round of review and you feel that this manuscript is now acceptable for publication, you may indicate that here to bypass the “Comments to the Author” section, enter your conflict of interest statement in the “Confidential to Editor” section, and submit your "Accept" recommendation.

Reviewer #1: All comments have been addressed

2. Is the manuscript technically sound, and do the data support the conclusions?

Reviewer #1: Yes

3. Has the statistical analysis been performed appropriately and rigorously? 

Reviewer #1: Yes

4. Have the authors made all data underlying the findings in their manuscript fully available?

Reviewer #1: Yes

5. Is the manuscript presented in an intelligible fashion and written in standard English?

Reviewer #1: Yes

6. Review Comments to the Author

Reviewer #1: Abstract

• I am a Statistician, but have never actually heard the term “bivariable multilevel logistic regression” before. What do the authors mean by this? Did you, perhaps, mean to say a univariate/unadjusted multilevel logistic regression?

• Delete were in the sentence “In this analysis, from the total 2405 participants, 14.3% ((95%CI: 12.1-16.8), (n=344)) of them were returned to health facilities for PNC use after they gave birth at a health facility”

Introduction

• Line from 54 to 56. Remained should probably be remains.

• Authors might wish to have someone check the grammar throughout the paper

Methods

• Great job explaining how the outcome variable was derived and how the independent variables were defined

• I would also like to commend the authors for explaining their choice of methods in detail

• Authors should avoid using the term bivariable/bivariate multilevel logistic regression. The appropriate term is univariate/unadjusted multilevel logistic regression.

Results

Socio-demographic and socio-economic characteristics

• Sentence 213: “were returned” should be returned. Can this be implemented throughout the manuscript, please

• Sentence 220: capitalize t on Table-1. Table should always have a capital T in the text

Associated factors with return to health facility for PNC (fixed effects)

• Odds ratios are supposed to be reported together with their reference categories. For instance, sentence 241: “Employed women were 51% (AOR=1.51, 95%CI: 1.04-2.19) more likely to return to health 242 facility for PNC during their postnatal period after they gave birth at health facilities” – more likely than who? The reference category should be mentioned for all the sentences explaining the odds ratios. This has been done for sentence “Women who delivered by caesarean section were also in higher odds (AOR= 2.53, 95%CI: 1.40-4.58) of returning to health facilities for PNC than their counterparts during their postnatal period.”, but not for the other sentences.

7. PLOS authors have the option to publish the peer review history of their article (what does this mean?). If published, this will include your full peer review and any attached files.

Reviewer #1: No

---

## [Author Response · Author response to Decision Letter 1]

22 Mar 2021

Dear editor and reviewer thank you very much for your comments and suggestions on the revised version of our manuscript. As usual, your comments and concerns are very important and critical to improve the manuscript accordingly. We tried to address the points in this re-revised manuscript. Here are the point by point responses!

Editor’s comments 

Please make sure that the manuscript is thoroughly edited for typographical and grammatical errors.

Authors ‘ response

Thank you very much for the concern. As much as possible, we thoroughly edit for typographical and grammatical errors. 

Reviewer’s comments 

Abstract

I am a Statistician, but have never actually heard the term “bi-variable multilevel logistic regression” before. What do the authors mean by this? Did you, perhaps, mean to say a univariate/unadjusted multilevel logistic regression?

Delete were in the sentence “In this analysis, from the total 2405 participants, 14.3% ((95%CI: 12.1-16.8), (n=344)) of them were returned to health facilities for PNC use after they gave birth at a health facility”

Authors ‘ response

Yes, we mean to “unadjusted multilevel logistic regression”. Dear reviewer it is common to say “bi-variable multilevel logistic regression” for unadjusted multilevel logistic regression considering the two variables (one dependent and one independent). However, to make it more clear we revised the manuscript according to your comments. Thank you very much! 

Thank you very much! 

Introduction

• Line from 54 to 56. Remained should probably be remains.

• Authors might wish to have someone check the grammar throughout the paper

Accepted! Thank you! 

Methods

• Great job explaining how the outcome variable was derived and how the independent variables were defined

• I would also like to commend the authors for explaining their choice of methods in detail

• Authors should avoid using the term bivariable/bivariate multilevel logistic regression. The appropriate term is

univariate/unadjusted multilevel logistic regression.

Thank you for your kind recognition and comments; your comments and concerns are considered. 

Results

Socio-demographic and socio-economic characteristics

• Sentence 213: “were returned” should be returned. Can this be implemented throughout the manuscript, please

• Sentence 220: capitalize t on Table-1. Table should always have a capital T in the text

Accepted! Thank you! 

Associated factors with return to health facility for PNC (fixed effects)

• Odds ratios are supposed to be reported together with their reference categories. For instance, sentence 241:

“Employed women were 51% (AOR=1.51, 95%CI: 1.04-2.19) more likely to return to health 242 facility for PNC during their postnatal period after they gave birth at health facilities” – more likely than who? The reference category should be mentioned for all the sentences explaining the odds ratios. This has been done for sentence “Women who delivered by caesarean section were also in higher odds (AOR= 2.53, 95%CI: 1.40-4.58) of returning to health facilities for PNC than their counterparts during their postnatal period.”, but not for the other sentences.

Thank you very much! Your concern is addressed in the re-revised version of the manuscript. 

Dear editor and reviewer, once again we thank you very much for all your valuable comments!

Kindly regards,

Brhane Ayele (on behalf of all authors)

---

## [Editor Report · Decision Letter 2]

25 Mar 2021

Do mothers who delivered at health facilities return to health facilities for postnatal care follow-up? A multilevel analysis of the 2016 Ethiopian Demographic and Health Survey.

PONE-D-20-18442R2

Dear Dr. Ayele,

We’re pleased to inform you that your manuscript has been judged scientifically suitable for publication and will be formally accepted for publication once it meets all outstanding technical requirements.

Kind regards,

Samson Gebremedhin, PhD

Academic Editor

PLOS ONE
---

## [Editor Report · Acceptance letter]

29 Mar 2021

PONE-D-20-18442R2 

Do mothers who delivered at health facilities return to health facilities for postnatal care follow-up?A multilevel analysis of the 2016 Ethiopian Demographic and Health Survey. 

Dear Dr. Ayele:

I'm pleased to inform you that your manuscript has been deemed suitable for publication in PLOS ONE. Congratulations! Your manuscript is now with our production department. 

Kind regards, 

on behalf of

Dr. Samson Gebremedhin 

Academic Editor

PLOS ONE